# Private Order Flows and Builder Bidding Dynamics: The Road to Monopoly in Ethereum's Block Building Market

## Abstract

Ethereum, as a representative of Web3, adopts a novel framework called Proposer Builder Separation (PBS) to prevent the centralization of block profits in the hands of institutional Ethereum stakers. Introducing builders to generate blocks based on public transactions, PBS aims to ensure that block profits are distributed among all stakers. Through the auction among builders, only one will win the block in each slot. Ideally, the equilibrium strategy of builders under public information would lead them to bid all block profits. However, builders are now capable of extracting profits from private order flows. In this paper, we explore the effect of PBS with private order flows. Specifically, we propose the asymmetry auction model of MEV-Boost auction. Moreover, we conduct empirical study on Ethereum blocks from January 2023 to May 2024. Our analysis indicates that private order flows contribute to 54.59% of the block value, indicating that different builders will build blocks with different valuations. Interestingly, we find that builders with more private order flows (i.e., higher block valuations) are more likely to win the block, while retain larger proportion of profits. In return, such builders will further attract more private order flows, resulting in a monopolistic market gradually. Our findings reveal that PBS in current stage is unable to balance the profit distribution, which just transits the centralization of block profits from institutional stakers to the monopolistic builder.

## CCS Concepts

• **General and reference** → **Empirical studies**; **Measurement**; **Evaluation**.

## Keywords

Ethereum, Builder market, Private Order Flow, Centralization, Monopoly

**ACM Reference Format:**
Anonymous Author(s). 2025. Private Order Flows and Builder Bidding Dynamics: The Road to Monopoly in Ethereum's Block Building Market. In *Proceedings of the ACM Web Conference 2025 (WWW'25)*. ACM, New York, NY, USA, 14 pages. https://doi.org/XXXXXXX.XXXXXXX

## 1 Introduction

Web3 represents a paradigm shift in online interactions, characterized by decentralized applications and services that leverage blockchain technology [56]. Ethereum, a key foundational layer for the Web3 ecosystem [61], is widely adopted for its censorship resistance and transparency, although it does not inherently ensure transaction privacy [59]. Before the final confirmation, transactions are sent to the public mempool, where they are visible to all nodes participating in the network. Miners then select, sequence, and bundle these transactions into blocks, which are added to the Ethereum blockchain. The dependence on transaction ordering within blocks creates space for arbitrage opportunities and even malicious activities such as frontrunning and sandwich attacks, resulting in financial losses for users [32]. The practice of manipulating the order of transactions is known as Miner Extractable Value (MEV) [15]. MEV not only causes user losses, but also poses a significant threat to the network. Intense competition among MEV searchers to exploit these opportunities can result in considerable network congestion and may even incentivize miners to reorganize the blockchain [11, 15].

In this context, an in-protocol mechanism for Proposer Builder Separation (PBS) has been devised to separate the roles of block construction and proposal [7, 24]. Within this framework, Builders are responsible for constructing blocks, whereas proposers are tasked with the proposal of blocks. In the initial design, builders were introduced to delegate the tasks of block construction and MEV extraction to specialized entities, enabling every Ethereum staker to participate as a proposer and earn rewards. After constructing the blocks, all builders engage in an auction to compete for their blocks to be selected on Ethereum. The proposer's sole responsibility is to choose the block offering the highest bid from builders. Ideally, when builders share common information, their equilibrium strategy would lead them to forgo all profits, similar to the dynamic between MEV searchers and miners before the introduction of PBS [45]. This mechanism ensures that rewards are distributed among all Ethereum stakers, preventing the concentration of MEV profits in the hands of institutional stakers [48].

However, empirical studies suggest that builders might acquire private transactions directly from wallets [54], which are not shared with all block builders, resulting in variations in block valuation [25]. Moreover, MEV searchers might direct their transaction bundles to selected builders. We refer to the bundles and transactions sent to the builders through these private channels as private order flows. As the builder market evolves, the share of private order flows has seen a substantial increase [5, 23]. This rise in private order flow introduces complexity to the landscape, as each builder now operates with a distinct transaction pool, leading to asymmetric competition among builders.

In this work, we explore whether PBS effectively achieves its intended objective of protecting the profits of all Ethereum stakers. The advent of private order flows changes the original transaction source of blocks, creates differences in the block valuation across different builders. We use *information difference* to elucidate this

in the block valuations. Our analysis, supported by empirical evidence, substantiates that information difference leads to different bidding strategies of builders, a phenomenon we refer to as *auction strategy difference*. It will cause builders to not bid all profits to Ethereum stakers and have different winning probabilities in the auction between builders and proposers. To make matters worse, our findings indicate that private order flows tend to favor the builder with higher winning probability. That exacerbates auction strategy difference and increases the probability of that builder with higher block valuation winning subsequent blocks and accumulating more private order flows. Eventually, the builder market becomes centralized. To verify this effect, we employ the framework of robust fairness [31]. The results demonstrate that existing differences compromise fairness within the builder market and culminate in a monopolistic condition. We then delineate that builders will retain more profits of such a monopolistic market, indicating greater losses of Ethereum stakers' profits.

Our primary contributions are:

- We identify two forms of differences in the builder market-information difference and auction strategy difference-and validate our theoretical analysis through bidding data from builders. Our findings indicate that private order flows, which induce information difference, account for up to 54.59% of the total block value. Furthermore, owing to auction strategy difference, the top 3 builders submit bids 26.87% lower than the other builders, while their total winning rate exceeds 95%. Our research reveals that, in the reality of information difference, the premise that builders bid all profits to proposers does not hold true.

- We analyze the impact of information difference and auction strategy difference on the builder market. Our finding suggests that the winning probability for builders with high-value blocks will continue to increase. We examine the robust fairness of the builder market, demonstrating that it fails to achieve robust fairness, inevitably leading to a monopolistic state. Extensive numerical experiments and on-chain data are used to validate our results. Our research reveals the monopolistic trend of the builder market from both theoretical and data perspectives.

- We investigate the implications of a monopolistic builder, including reduced earnings for proposers and increased discrimination in block construction. Our results reveal that within a more monopolistic builder market, the maximum average profit margin attained by builders is 27.66% and the delay gap between transactions with lower and higher priority fees expands nearly 16 times. It means that the monopolistic builder market will further deviate from PBS's original idea of protecting Ethereum stakers profits.

## 2 Background

In this section, we present necessary background to facilitate a clear comprehension of our paper.

### 2.1 Proposer Builder Separation.

The Proposer Builder Separation (PBS) mechanism has been suggested as a critical innovation in Ethereum [20]. Subsequent to

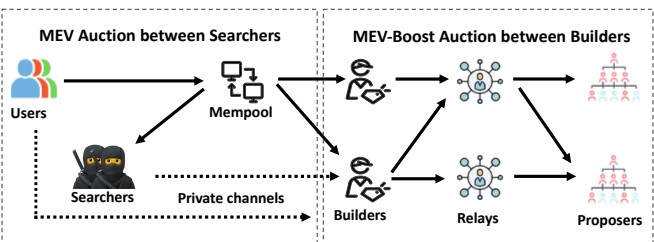

**Figure 1: Two-phase auction in PBS.**

November 2022, PBS has been instrumental in the formulation of nearly 90% of the blocks, and the top three builders have more than 70% of the market share [55]. PBS delineates the roles that were formerly consolidated in miners, segregating them into block builders and block proposers [7]. This separation serves a dual purpose. Firstly, it mitigates centralization apprehensions arising from the economies of scale linked to MEV extraction by proposers within PoS. It will balance the distribution of MEV profits, rather than concentrate in the institutional stakers. Secondly, it acts as a deterrent against MEV extraction by proposers and preserves preconfirmation confidentiality, as proposers are restricted to accessing only the block header prior to its finalization [8].

Due to the challenges associated with achieving compatibility at the consensus layer, Flashbots has introduced MEV-Boost [55], an off-chain approach to implement PBS as an interim solution. Figure 1 delineates an architectural representation of the MEV auction and MEV-Boost infrastructure, which constitutes a two-phase auction mechanism encompassing four distinct roles: searcher, builder, relay, and proposer.

Phase One: MEV auction between searchers. The MEV auction entails the auctioning of block space packaged by block builders, where the priority fee is given to the latter. Initially, searchers bundle their transactions with victim transactions, transmitting them to builders through a private channel [13, 46]. In particular, searchers are relieved of the obligation to pay gas fees for unsuccessful bundles, thereby mitigating the associated risks of MEV extraction. This characteristic differentiates it from the partial all-pay auction format employed in previous English auctions [21, 58, 68]. Victim transactions within the bundle predominantly originate from the public mempool, with the extractable profit being transparently calculable. Consequently, for searchers, the utility of a bundle is contingent not only on its inclusion but also on its positional prioritization [2, 40].

Phase Two: MEV-Boost auction between builders. MEV-Boost auction constitutes a bidding mechanism to secure the opportunity to finalize blocks. Transactions are typically prioritized based on the associated priority fees or the builder's profit maximization strategy, thus optimizing revenue within the constrained block space [1]. Subsequently, the blocks are submitted to the relays. Throughout the interval, builders recurrently execute the sorting algorithm and persistently submit blocks. This iterative process stems from the asynchronous nature of the network, which produces an unpredictable deadline for the auction within the slot, despite the fixed slot duration of 12 seconds [4]. Relays identify the

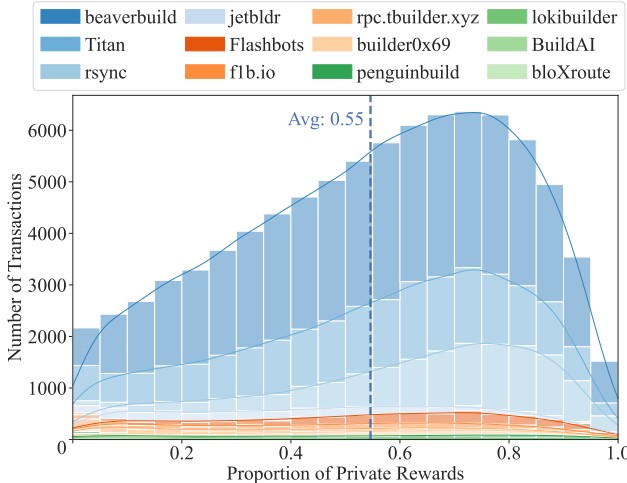

Figure 2: Proportion distribution of private rewards among builders.

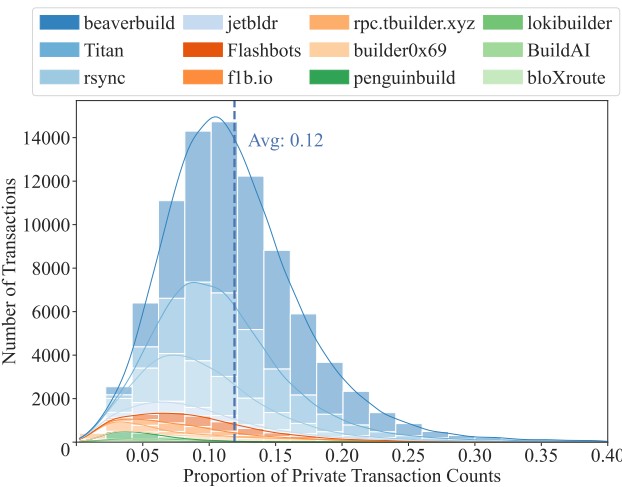

Figure 3: Proportion distribution of private transaction counts among builders.

most profitable block and send its header to the designated proposer for validation [14]. In the final stage, the proposer endorses the highest bidding block for the finalization of the blockchain [44, 48].

## 2.2 Robust Fairness.

Robust fairness denotes that the stochastic outcome of a block builder's reward $\lambda$ is aligned with its initial investment $\lambda_0$ [31]. For any given pair of parameters $(\varepsilon, \delta)$ such that $\varepsilon \geq 0$ and $0 \leq \delta \leq 1$, an incentive mechanism preserves a $(\varepsilon, \delta)$-fairness for miner $A$ possessing a fraction $a$ of the total resource if $A$ receives a fraction $\lambda$ of the total reward satisfying

$$\Pr\left[(1-\varepsilon)\lambda_0 \leq \lambda \leq (1+\varepsilon)\lambda_0\right] \geq 1 - \delta. \tag{1}$$

According to this definition, diminished values of $\epsilon$ and/or $\delta$ indicate higher levels of fairness. Note that $\lambda$ will gradually converge as long as the number of auction rounds increases. Robust fairness is articulated through $(\epsilon, \delta)$-fairness, which delineates the extent of robust fairness. In subsequent sections, this definition will be used to evaluate the dynamic of concentration in the builder network.

## 3 Differences in PBS

Within the MEV-Boost framework, certain transactions and bundles are recognized to bypass the public mempool, instead being directed to builders, which is called private order flows [25]. The influence of private order flows on the valuation of blocks is substantial [18]. This section quantifies the impacts of private order flows and characterizes them as information difference. Under the conditions of information difference, builders with disparate block valuations will develop different bidding strategies. A model for the MEV-Boost auction has been established, demonstrating that different builders employ varying bidding ratios and exhibit different winning probabilities. Moreover, we employ empirical data to substantiate our theoretical exposition.

### 3.1 Dataset

First, we deploy an Erigon node and a Lighthouse node to sync the block and transaction information. Using Flashbots Mempool Dumpster [22], we gather transaction data from the public mempool from November 2023 to May 2024. We filter the transactions that have appeared in the public mempool from the on-chain transaction dataset to obtain the private transaction dataset. Second, we build the MEV-Boost auction dataset through interfacing with various relay APIs. We acquire all bidding processes by utilizing the 'builder blocks received' and 'proposer payload delivered' endpoints of public APIs from relays. Our datasets range from January 2023 to May 2024.

### 3.2 Information Difference

In practical scenarios, emergent and weak builders entering the network often encounter difficulties in acquiring order flows from searchers and users. This is attributed to the preference of searchers and users for routing their bundles and private transactions to builders boasting significant market shares, thereby enhancing the likelihood of being selected on Ethereum. It implies that the valuation of the blocks constructed by different builders will vary significantly due to the integration of private order flows. We refer to this phenomenon as the *information difference* between builders. We plot the proportion of private rewards for builder in Figure 2, which uses distinct colors to represent different builders, arranged in descending order based on the market share spanning from block 19,331,051 to 19,431,051. Figure 3 illustrates the proportion distribution of private transaction counts among builders. The solid line denotes the counts of private transactions for builders. The chart indicates that builders have higher proportions of private transactions and private rewards compared to others with smaller market shares. The results show that, despite constituting only 12% of the total amount of transactions, private order flows significantly contribute to 54.59% of block rewards.

We examine two types of representative builders and impose the following assumptions within the builder market. In the context of the MEV-Boost auction, the builders that exert influence on the auction outcome are predominantly the first and second highest ranked builders in the valuations and bidding of the blocks, respectively [62]. We assume there are two builders, i.e., builder $\mathcal{P}_i$ and builder $\mathcal{P}_j$ are competing for building blocks.

Initially, the private order flows of builder $\mathcal{P}_i$ (resp. $\mathcal{P}_j$) is represented as $a$ (resp. $b$). In particular, considering the private order flows are possible to be submitted to multiple builders, we recomputed the overlapping private order flows and utilized $a + b$ as the total counts within the market. The expectation counts of the private order flows of each builder's respective block satisfy a binomial distribution. Therefore, for builder $\mathcal{P}_i$ (resp. $\mathcal{P}_j$), the proportion of private order flows is represented as $\frac{a}{a+b}$ (resp. $\frac{b}{a+b}$).

We now focus on the dynamic changes of the private order flows of builders. Due to high subsidies, rsync-builder has grown from an emerging weak builder in January 2023 to a strong builder occupying the third market share [17]. Therefore, studying the data of the rsync-builder will provide important insights into the dynamic changes. we have calculated the quantity of private bundles received by the rsync-builder over a five-month period, spanning from its inception to its attainment of a top-three position within the network in 2023, as depicted in Figure 4. The red and deep blue lines represent the volume of private bundles sent by all searchers and the top five searchers, respectively. It is apparent that upon its initial entry into the network, the rsync-builder attracted a minimal volume of private bundles. However, as the winning probability of the rsync-builder expanded, it increasingly attracted private bundles, increasing from nearly 0 to in excess of 15,000 by May. What's more, when rsync's market share decreases, the number of private bundles it receives will also decrease. Therefore, we find there is a positive correlation between the number of searcher connections and the winning probability of builders.

What's more, upon analysis of decentralized finance protocols like MEV Blocker and BackRunMe, known for delivering large volumes of user private order flows, it becomes evident that these entities consistently engage in collaboration with strong builders [6, 16]. This means that both private bundles from searchers and private transactions from users tend to be sent to builders with higher winning probabilities. In light of these findings, we make the following assumption.

ASSUMPTION 1. *Let $Z_t^i$ denote the proportion of private order flow connected to builder $\mathcal{P}_i$ at round $t$. $Z_0^i = \frac{a}{a+b}$ and $Z_t^i \in [0, 1]$ for all $t \geq 0$. We assume that $\{Z_t^i\}_{t \geq 0}$ is a stochastic process with the following dynamics: Before each round $t$, private order flows with a quantity of $\delta_t$ will choose to connect to the previous winner of the MEV-Boost auction, while they have a probability of $p_t \in [0, 1]$ to drop the connection of the previous failed builder.*

Here, $\delta_t$ is non-negative value representing the changes in private order flows at round $t$. When $\delta_t$ is 0, the private order flows are unchanged. For builder $\mathcal{P}_j$, we represent it as $Z_t^j$, where $Z_t^j = 1 - Z_t^i$ for all $t \geq 0$.

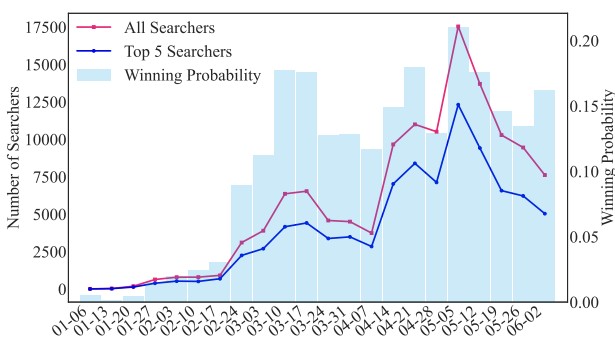

**Figure 4: Weekly distribution of connected searchers since the emergence of rsync-builder.**

## 3.3 Auction Strategy Difference

The presence of information difference leads to different bidding strategies among builders participating in the MEV-Boost auction within the PBS framework. This section predominantly elucidates the derivation of auction strategy differences between builders and corroborates these findings with empirical data.

We define the MEV-Boost auction as $A(\mathcal{P}, \mathcal{V}, \mathcal{S})$, representing participants, block valuation, and bidding strategies, respectively. For block valuation $v_t^k$ of builder $\mathcal{P}_k, k \in \{i, j\}$ at round $t$, we have the following relationship as

$$\Delta_t^k = N_t Z_t^k w_t^k, \quad (2)$$

$$v_t^k = g(\Delta_t^k, r_t). \quad (3)$$

In this context, $\Delta_t^k$ represents the total value of private order flows that builder $\mathcal{P}_k$ can build in round $t$. $N_t$ represents the total number of private order flows and $w_t$ is the average profit provided by each private order flow. The auction floor price $r_t$ is set to the valuation of the block that can be constructed from public transactions. From our observation in Section 3.2, the block valuation $v_t^k$ is an increasing function of $Z_t^k$. The builder $\mathcal{P}_i$ (resp. $\mathcal{P}_j$) uses the bidding function $s_t^i(v_t^i, r_t)$, where $r_t$ also denotes the reservation price of the proposers. Consequently, the bidding prices can be expressed as $b_t^i = s_t^i(v_t^i, r_t)$ (resp. $b_t^j = s_t^j(v_t^j, r_t)$). We denote that the valuation of the block $v_t^k$ has its cumulative distribution function $F_k(\cdot)$. With quasilinear utility function , we can derive builder $\mathcal{P}_i$'s utility function as follows

$$u_i(v_i, b_i, b_j) = \begin{cases} v_i - b_i, & \text{if } b_i \geq b_j, b_i \geq r, \\ 0, & \text{otherwise.} \end{cases} \quad (4)$$

Builder $\mathcal{P}_i$ chooses its bidding price $b_i$ by maximizing

$$R(v_i, b_i, b_j) = (v_i - b_i) F_i[s_j^{-1}(b_i, r)]. \quad (5)$$

There exists a solution to the following pair of differential equations

$$\begin{cases} -F_i\left[s_j^{-1}(b_i, r)\right] + \frac{(v_i - b_i) F_i'\left[s_j^{-1}(b_i, r)\right]}{\frac{\partial}{\partial v_j} s_j\left[s_j^{-1}(b_i, r), r\right]} = 0, \\ -F_j\left[s_i^{-1}(b_j, r)\right] + \frac{(v_j - b_j) F_j'\left[s_i^{-1}(b_j, r)\right]}{\frac{\partial}{\partial v_i} s_i\left[s_i^{-1}(b_j, r), r\right]} = 0. \end{cases} \quad (6)$$

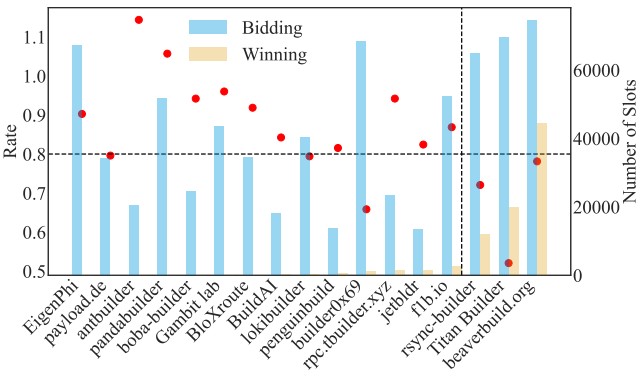

**Figure 5: Bidding strategies and winning rate of builders.**

THEOREM 1. *Strong builder $\mathcal{P}_i$ bids less aggressively than Weak builder $\mathcal{P}_j$ for each valuation $v$ while being more likely to win in a single round.*

The proof of Theorem 1 can be found in Appendix C. Consequently, Theorem 1 means that the builder with a higher valuation distribution is more likely to submit higher bids, even though his bidding strategy tends to be more conservative. We refer to this phenomenon as the auction asymmetry between builders. We also find the evidence in real-world data.

To present empirical evidence, we utilize builder bidding data on 100,000 blocks from the end of February to mid-March 2024 to analyze the participation of various builders in the MEV-Boost auction and their success rates, as shown in Figure 5. Furthermore, we compute the ratio of successful auction bids to the block value to validate Theorem 1. The right axis of Figure 5 illustrates the number of slots. The blue and yellow bars depicted in the figure denote the frequency of the builder's participation in the MEV-Boost auction and their respective winning counts. The left axis of Figure 5 represents the ratio of successful auction bids to the block value. The average bid proportion for each builder is indicated by red dots. Moreover, a dashed line parallel to the x-axis separates the left axis at a ratio of 0.8, and we use a dashed line perpendicular to the x-axis to categorize the builders into the top 3 and other builders. Notably, the top 3 builders' blocks constitute over 90% of all network blocks. The bidding proportions of the top three builders are markedly below 0.8, significantly lower than those of lower-ranked builders. However, the MEV-Boost auction rounds they win exceed those with substantially higher bidding proportions. This evidence precisely aligns with our previous Theorem 1.

## 4 Centralization in Builder Market

With information difference and auction strategy difference, the dynamic impact on private order flows and builder winning probability is worth to be further studied. This section investigates changes in market concentration among builders by employing the framework of robust fairness. By our theorem, more and more private order flows are concentrated on the winning builder, consequently leading to a monopoly in the builder market. Our theorem is supported by numerical simulations and empirical data.

### 4.1 Robust Fairness

In our paper, the robust fairness is used to evaluate whether the random outcome of a builder's winning probability will be concentrated on its initial value. What's more, in order to study the dynamic changes of private order flows and the winning probability of builders, our analysis utilizes the techniques of Stochastic Approximation (SA) [47, 49]. The formal definition and associated lemmas of SA are presented in Appendix C.

We establish that $\{Z_t^i\}$ is an SA algorithm. In particular, the update of $Z_t^i$ is driven by the winning probability $\lambda_i$ of builder $\mathcal{P}_i$ in the succeeding MEV-Boost auction, denoted by $f(\cdot)$. Subsequently, we use the SA algorithm to investigate the asymptotic properties of $Z_t^i$. Specifically, we discover that $Z_t^i$ will almost surely converge to either 0 or 1, which indicates that builder market cannot achieve robust fairness.

THEOREM 2. *As rounds $n$ approach infinity, the winning probability $\lambda_i$ of builder $\mathcal{P}_i$ winning converges to 0 or 1 with certainty. Consequently, the PBS builder market fails to achieve robust fairness, ultimately resulting in a monopolistic state.*

The comprehensive proof is detailed in Appendix C. Theorem 2 states that the winning probability of the builder will converge to 0 or 1. The builder with a winning probability of 0 will exit the network successively due to running costs. Ultimately, only one builder will carry out block construction and the builder market becomes monopolized.

### 4.2 Experimental Evaluation

In this section, we assess the fairness of PBS under different values $p_t$ using numerical simulations. In our experimental evaluation, the quantity of order flows $N_t$ within a single slot follows a Poisson distribution. The average profit $w_t$ of each private order flow is represented as a random variable drawn from a log-normal distribution, which aligns with the previous study of the MEV-Boost auction [60]. In addition, we have also performed an analysis of victim transaction counts within each block, covering a range of 100,000 blocks from block 18,966,775 to block 19,066,775. The results of the chi-square test reveal that these counts conform to a Poisson distribution.

For the purpose of robust fairness assessment, the parameters are predefined as $\epsilon = 0.1$ and $\delta = 10\%$. Consequently, there exists a probability of at least 90% that the return on winning probability for a builder, given a stochastic outcome, falls within [0.9, 1.1] of its initial private order flow proportion. For practical implementation, the interval $[(1 - \epsilon)\lambda_0, (1 + \epsilon)\lambda_0]$ is designated as the *fair area*, with the lower and upper bounds of the simulation domain corresponding to the 5th and 95th percentiles, respectively.

Figure 6 depicts the evolution of $\lambda_i$ along with the 6,000 rounds under $Z_0^i = 0.6, \delta_t = 0.0002, w_t \sim$ Log-normal$(0, 1)$ and $N_t \sim$ Poisson(5). The fair area is set to [0.54, 0.66], which captures the robust fairness for block builders with information asymmetry and auction asymmetry. In the numerical simulation, we examined three scenarios characterized by $p_t = 0$, $p_t = 0.5$ and $p_t = 1$. Consistent with Theorem 1, we posit that the stronger builder will offer 70% of the block value as their bid, while the weaker builder will offer 90% of the block value to the proposer. Initially, builder $\mathcal{P}_i$ wins 300 of

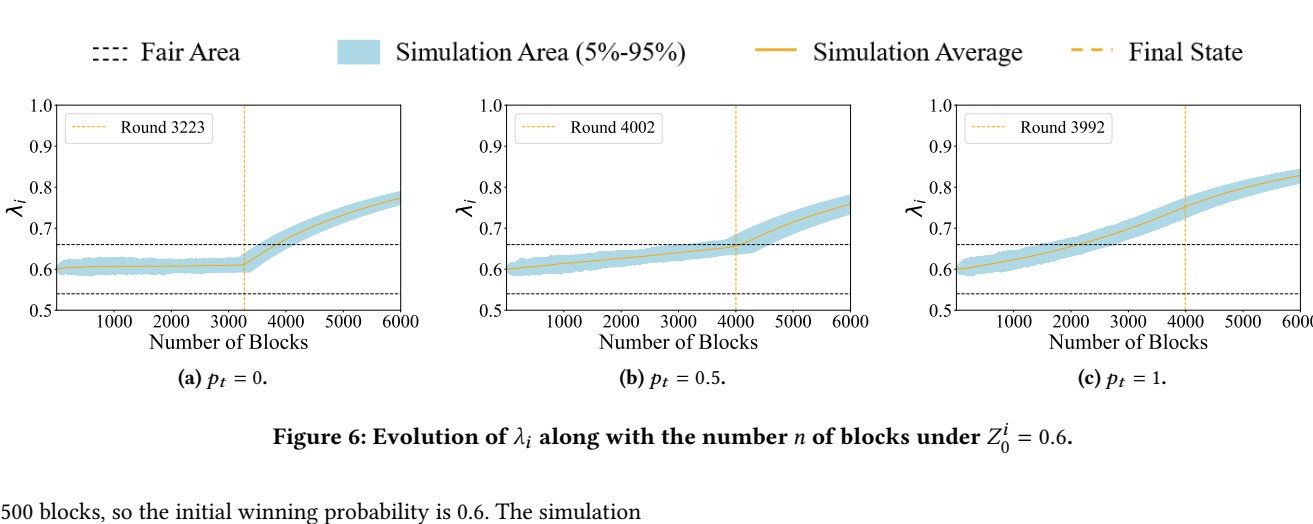

Figure 6: Evolution of $\lambda_i$ along with the number $n$ of blocks under $Z_0^i = 0.6$.

500 blocks, so the initial winning probability is 0.6. The simulation tracks the winning probability $\lambda_i$ of builder $\mathcal{P}_i$ after 6,000 blocks across different scenarios. We repeat the simulations 1,000 times and report the statistical results.

Figure 6 displays the variation of $\lambda_i$ with an increasing number of blocks for different $p_t$. It is evident that $\lambda_i$ will continue to increase, with a more pronounced increase after block 3,223, block 4,002 and block 3,992 respectively, since at that round $Z_t^i = 1$. After that, the builder $\mathcal{P}_i$ will win all subsequent MEV-Boost auctions. In Figure 6a, we observed a maintenance phase in the fair areas. $\lambda_i$ is exhibiting a slow upward trend at a gradual pace. The reason is that when $p_t = 0$, many private order flows become public information due to the simultaneous connection of two builders.

The scope of the experiment has been extended to include the scenario of multiple builders, as detailed in Appendix B. Our findings suggest that the presence of multiple builders does not alter the inherent market tendency toward monopoly. Furthermore, our research also concerns the collaboration between searchers and builders, as well as the dynamics of timing games. A detailed analysis of these aspects is provided in Appendix A. In the context of searcher builder collaboration, should the weaker builder establish a fixed partnership to supply private order flow searchers, the market will not advance towards a complete monopoly. However, for this set of searchers, collaboration with the weaker builder is not the optimal strategy for maximizing profits. Regarding the proposer timing game, while the ultimate monopoly outcome remains unchanged, the duration of the transition from oligopoly to monopoly is altered.

The observation aligns with Theorem 2 that PBS cannot achieve robust fairness. For builders with information and auction asymmetry, the emergence of a monopoly state is inevitable. In conjunction with numerical simulations, we scrutinize block data generated within the PBS framework to furnish more evidence that supports our simulations. Figure 7 delineates the variations in network concentration among block builders from September 2023 to mid-May 2024. We compute a weekly average of the builder's market share and employ the Herfindahl-Hirschman Index (HHI), as utilized in [26], to evaluate the network concentration over time. Empirical data elucidate a progressive intensification in the concentration of the builder market, with the market share of the top-ranked builder increasing from 18% to nearly 50%. This trend also indicates

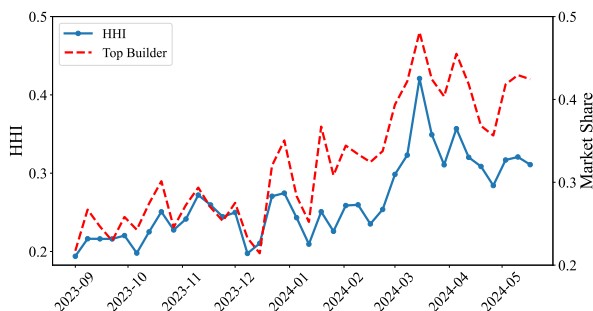

Figure 7: The network concentration.

the road to a monopoly state within the builder market based on real-world data.

## 5 Implications of Drifting to Monopoly

In this section, we examine the adverse consequences in the transition process toward a monopolistic builder market. This includes proposer revenue reduction and block construction discrimination. The former will result in the builder retaining a greater portion of the rewards and sending fewer rewards to the proposer. The latter can lead to longer transaction delays, especially for transactions that offer fewer priority fees.

### 5.1 Proposer Revenue Reduction

In the MEV-Boost auction, it can be mathematically demonstrated that builder $\mathcal{P}_i$ exhibits first-order stochastic dominance over builder $\mathcal{P}_j$ [12]. However, in a surplus equivalence symmetric setting [9], where the builder $\mathcal{P}_i$ competes with another builder with a similar distribution, the bid approach of the builder $\mathcal{P}_i$ tends to be less aggressive in the asymmetric case, which will reduce the profits of stakers.

Given the inverse bidding functions, it is denoted as $\phi_s(b)$ in the symmetric situation and $\phi_a(b)$ in the asymmetric situation. We can formally describe this bidding difference using the following

**Table 1: The market share and profit margin of builders.**

| Builders | Market Share [%] | Profit Margin [%] |
|----------|------------------|-------------------|
| beaverbuild | 50.5 | 13.58 |
| Titan | 22.56 | 8.34 |
| rsync-builder | 13.64 | 27.66 |
| jetbldr | 3.09 | -9.91 |
| Flashbots | 2.89 | 12.34 |
| f1b | 1.80 | 13.48 |
| tbuilder | 1.52 | -5.77 |
| builder0x69 | 1.49 | 16.68 |
| penguinbuild | 0.80 | 14.47 |
| lokibuilder | 0.41 | 13.47 |

equation which is

$$F_s\left[\phi_s(b)\right] < F_a\left[\phi_a(b)\right]. \tag{7}$$

$$\phi_a(b) < \phi_s(b). \tag{8}$$

To illustrate the relationship between the level of asymmetry and the loss of income for proposers, we consider $K$ builders in the PBS framework, each with i.i.d. valuations. Their cumulative distribution function, denoted by $H(v)$, is assumed to be continuously differentiable. These builders are envisioned to form cartel organizations, assuming two cartels: $m + n = m' + n' = K$, where $m > n$ and $m' > n'$. This framework can be extended to involve multiple entities.

In scenario $I$, the distribution functions are $F_1^I(v) = H(v)^m$ and $F_2^I(v) = H(v)^n$, leading to equilibrium $(\phi_1(b), \phi_2(b))$. In contrast, in scenario $II$, the distributions are $F_1^{II}(v) = H(v)^{m'}$ and $F_2^{II}(v) = H(v)^{n'}$, resulting in equilibrium $(\widetilde{\phi}_1(b), \widetilde{\phi}_2(b))$. The expected revenue of the proposers in situation $I$ is denoted as $R^I$ and the expected revenue of proposers in situation $II$ is denoted as $R^{II}$. If $m < m'$, situation $II$ is more asymmetric than situation $I$, and we have $R^I > R^{II}$. That is

$$R^I = \int b \, d\left(H\left[\phi_1(b)\right]^m H\left[\phi_2(b)\right]^n\right),$$

$$R^{II} = \int b \, d\left(H\left[\widetilde{\phi}_1(b)\right]^{m'} H\left[\widetilde{\phi}_2(b)\right]^{n'}\right).$$

THEOREM 3. *The inherent difference significantly decreases the income of proposers compared to a symmetric auction. Furthermore, an increase in the level of difference notably exacerbates the phenomenon of income loss.*

Note that a more detailed proof of Theorem 3 can be found in the Appendix C. The information shown in Table 1 presents the profit margins and market shares of the top ten builders in March 2024 during the MEV-Boost auctions. In order to compare with builder markets with different levels of concentration, we adopted the same profit margin calculation method from previous research [42]. In our timeframe, the average HHI index is 0.35 from February 29 to March 15, 2024. The builder named beaverbuild constructs more than half of the blocks and has a profit margin of approximately 13.58% per block, while rsync-builder possesses 13.64% of the market and averages a profit of 27.66% per block. In contrast, in previous research, the average HHI index is 0.23 from October 2023 to March

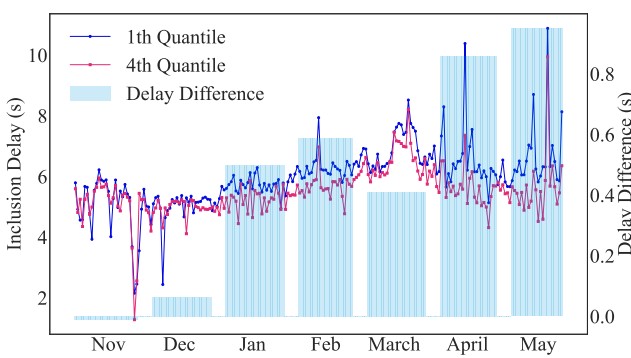

**Figure 8: Transactions inclusion delay in public mempool.**

2024 [42], which means that the builder market is less concentrated. The highest profit margin is only 5.4%, substantially lower than our results. Our findings corroborate Theorem 3, which posits that auction strategy difference in builders' competition exacerbates revenue loss for proposers. Additional detailed results, including total profits and payments to the proposers for each builder, are available in Appendix D.

## 5.2 Block Construction Discrimination

As per our analysis in Section 3, with the intensification of the information difference, Strong builder can obtain a higher value of the private order flow. Weak builders tend to withdraw more transactions from the public memory pool in order to compensate for the private order flow value obtained through their private order flow and the gap with Strong builders. And Strong builder, since it has more private order flow, withdraws less data from the public mempool. To put it differently, an increase in the disparity of block evaluations results in a prolonged average waiting period for regular transactions, especially for those with lower priority fees, within the public mempool.

We examined the transaction delays in the public mempool during the period from November 2023 to May 2024, as shown in Figure 8. This interval is calculated by subtracting the time that transactions enter the public mempool from their actual inclusion time. We rank priority fees from lowest to highest and analyze the mean daily latency for transactions in both the highest and lowest quartiles. We select transaction delays within the range of 5% to 95% to mitigate the impact of extreme values.

The blue line shows the delay for transactions with a higher priority fee corresponding to the 25% quartile, while the red line illustrates the transaction delay for lower priority fees at the 75% quartile. It is observed that the lower the priority fee paid, the longer the transaction delay. The light blue bar depicts the average monthly delay difference between these two categories of transactions. As beaverbuild's market share has been steadily growing since January, the concentration within the network has increased, leading to a growing delay gap between transactions with lower and higher priority fees. This gap has widened nearly 16 times, from 0.06 to 0.95. These results are consistent with our previous deductions.

## 5.3 After Reaching Monopoly

In the presence of a monopolistic builder, the interaction between the builder and the proposers will be subject to stringent unfair conditions. This scenario is analogous to a *dictator game*. Based on previous behavioral experiments [19], it can be deduced that as the game continues to repeat, the monopolistic builder is likely to become more selfish and eager to intercept all profits. In other words, in a fully competitive builder market, the majority of these profits are directed towards the proposer. In contrast, following the monopoly of the builder market, the profit of the proposers will be adversely affected. In the most extreme scenario, the proposers can only receive the block floor price and cannot earn any additional profits from private order flows.

Moreover, there is also the risk that the monopolistic builder conditionally excludes specific transactions. In the PBS architecture before 2023, there is evidence to support that the four largest builders (Flashbots, Builder0x69, BloXroute, and beaverbuild) will all censor the blocks. Their blocks are observed to omit transactions associated with Tornado Cash, such as deposits to and withdrawals from the Tornado Cash contract [53]. In 2024, 60% builders will still review transactions in blocks [52]. This practice also means that the builder can selectively exclude transactions, especially in the monopoly case. The monopolistic builder can decide which transactions to exclude from each block. Provided that the exclusion of these transactions does not cause the block value to fall below the auction floor price, the monopolistic builder can exclude these transactions without any impact of the auction result.

## 6 Related Work

**Miner Extractable Value.** The concept of Miner Extractable Value (MEV) was originally introduced in Flash Boys 2.0 [15], denoting the potential profits that miners can accrue by reordering transactions within smart contracts. The primary objective of this research was to quantify and detect MEV, which constitutes a fundamental component of blockchain technology analysis. Qin et al. [45] systematically quantified a range of tactics to extract MEV, including front-running, back-running, and sandwich attacks. Notably, within a sandwich attack scenario, the use of automated market maker mechanisms in decentralized exchanges induces deterministic price alterations based on transactional directionality [3, 66, 67]. Heimbach et al. [28] elucidated that MEV could substantially erode user profits. This deterministic characteristic facilitates adversarial prediction and exploitation of transaction outcomes. Recently, the concept of Non-atomic Arbitrage was introduced in [27], demonstrating how searchers capitalize on disparities between centralized and decentralized exchanges. Furthermore, Wang et al. [57] identified cyclic arbitrage as an innovative MEV strategy. Li et al. [35] executed an exhaustive investigation of Flashbots bundles, discovering 17 novel DeFi MEV strategies. The research on MEV has also been extended to the NFT field [39, 65]. These findings illuminate the intricate and evolving nature of MEV methodologies. Our paper is pioneering in measuring MEV impacts within PBS and delineate associated risks. This analytical endeavor is crucial for understanding the extensive repercussions of MEV and PBS in Ethereum.

**Proposer Builder Seperation.** The PBS ecosystem has been rigorously investigated in previous scholarly endeavors, yielding profound insights into its dominant dynamics and future developments [10, 42, 53, 62, 63]. Heimbach et al. [26] delineated an unequivocal depiction of burgeoning centralization, particularly within the builder and relay sectors of the PBS ecosystem, illuminating the substantial control exerted by a few dominant entities. Similarly, Wahrstatter et al. [54] performed a comprehensive analysis of the competitive builder market, elucidating the implications of vertical integration in diverting value and further consolidating power. The work in [25] examined the implications of private order flow auctions on the PBS equilibrium. It was contended that these transformations not only disrupted the current framework but also rendered the ecosystem more susceptible to vulnerabilities. Our investigation uniquely forecasts the strategic maneuvers of builders that might result in not only centralization but also a monopolistic state, examining the intricate and delicate interrelationship in the builder market, thereby highlighting concerns regarding fairness.

**Frontier to Ethereum.** The majority of existing studies focus on the frontier study of blockchain, including smart contracts [34], fraud behaviors [29] and wash trading detection [51]. Lin et al. [36] introduced a novel money laundering detection algorithm, while Li et al [33] researched on the phishing scams. Huang et al. [30] measured the prosperous NFT ecosystem and revealed the facade of decentralization. The use of NFT arbitage and airdrop is carried out in [65]. Additionally, Wu et al. [61] designed a transaction semantic extraction method, and Zhao et al. [64] performed a network analysis and identified some anomaly behaviors in Ethereum.

## 7 Conclusion

In this paper, we identify the information difference among builders and derive the auction strategy difference in MEV-Boost auction. The analysis indicates that private transactions contribute significantly to the information difference among builders. This produces divergent valuations of blocks and different bidding strategies during the MEV-Boost auction stage, with the dominant builder showing a bidding ratio of 26.87% lower than that of other builders. Our findings suggest that private order flows preferentially benefit the builder with a higher winning probability. This tendency accentuates the difference in auction strategies and enhances the winning probability of the said builder. To verify this effect, we employ the framework of robust fairness. Our model demonstrates that existing differences compromise fairness within the builder market and culminate in a monopolistic condition. Then we highlight significant concerns posed by monopolistic builders, including issues related to profit distribution and transaction discrimination. . These insights are essential for the assessment of the current PBS mechanism and the future development of more equitable mechanisms within the blockchain ecosystem.

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

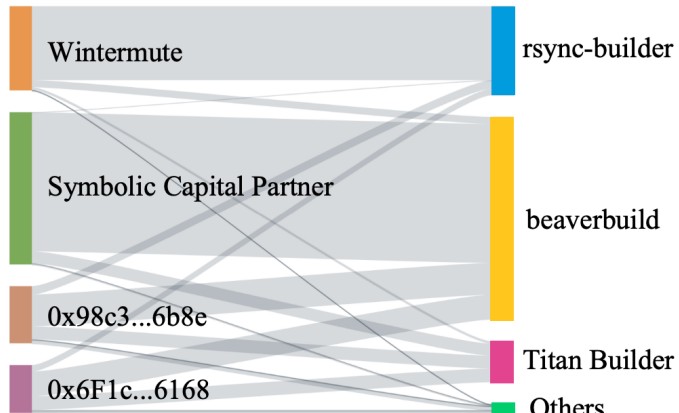

**Figure 9: Evidence of searchers-builders collaboration.**

## A Extension to Different Dynamics

We extend our analysis to include two more intricate scenarios. Our findings indicate that in the scenario of collaboration between builders and searchers, there is a low-level robust fairness. Additionally, the proposer timing game introduces enhanced winning opportunities for select builders, thereby altering the time required to achieve monopoly.

*A.0.1 Searcher Builder Collaboration.* We commence by examining the collaboration between builders and searchers. This entails searchers exclusively providing bundles for a particular builder, regardless of their success or failure in previous builder auctions. As illustrated in Figure 9, this collaboration is evident. We can contrast two sets of searchers: Wintermute & Symbolic Capital Partner and 0x98c3 & 0x6F1c. Standard searchers distribute their bundles across multiple builders, thus enhancing their chances of inclusion on the blockchain. The 0x98c3 & 0x6F1c group exemplifies this approach. Their bundles are similarly associated with various builders. However, the Wintermute and Symbolic Capital Partner operate differently. More than 95% of their bundles are sent to rsync-builder and beaverbuild, respectively. This highlights a collaboration between Wintermute and rsync-builder, as well as Symbolic Capital Partner and beaverbuild.

The collaboration between searchers and builders can lead to private order flows even if some builders have low winning probabilities. However, it should be noted that the searcher who cooperates with builder $\mathcal{P}_i$ did not actually employ the optimal strategy to maximize profits. Because they cannot get the maximum chance of being on the blockchain. The simulation result is shown in Appendix B.

*A.0.2 Timing Game.* Another scenario involves the proposer delay, referred to as the timing game between relays and proposers [50]. In essence, this theory elucidates that proposers aim to prolong the process of selecting builders' bids as much as possible, thereby encompassing block value within the extended timeframe. Following the public disclosure of the timing game strategy on the proposer named p2p.org [43], an increasing number of builders and proposers are engaging in the timing game to earn greater block profits. Real-world data indicates that proposers are willing to delay the time to

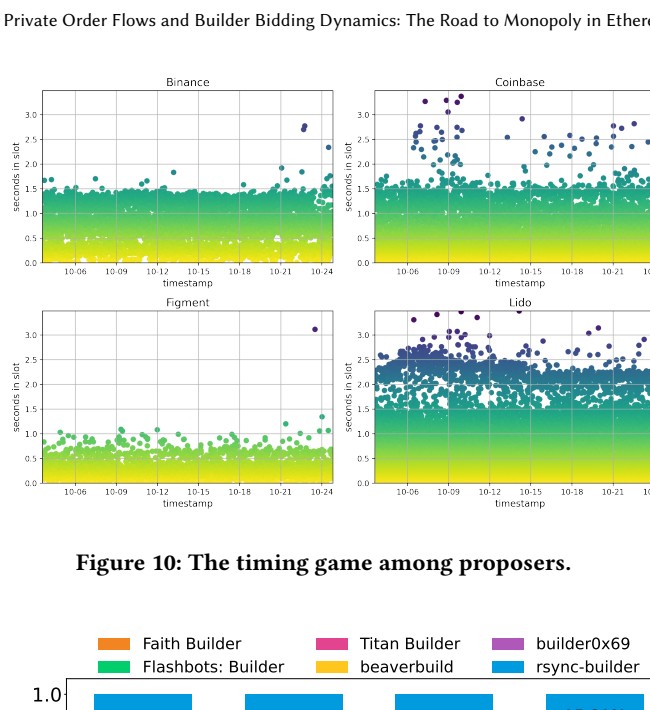

**Figure 10: The timing game among proposers.**

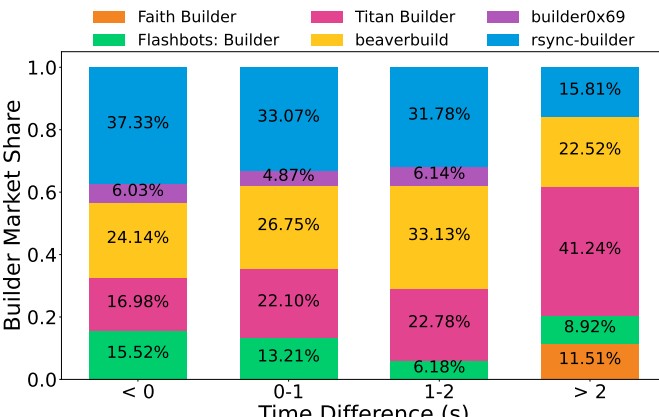

**Figure 11: Impacts of winning bid selection time on winning probability.**

get the block header [41], and Figure 10 illustrates the existence of this phenomenon. Lido and Coinbase hold more than 45 % of the market share, but many of their chosen slots come from 2 seconds later, as a comparison Figment, which is third in market share, has almost all of its slots coming from 1.5 seconds earlier.

Figure 11 reveals the impacts of the timing game for different builders. The data spans from slot 7,460,000 to 7,620,000, and we analyze the top five builders in terms of market share during that period. Evidently, if the selection time for the winning bid exceeds 2 seconds, the likelihood of Titan Builder winning increases by approximately 20%, while the probability of rsync-builder winning decreases correspondingly. In other words, the timing game also affects the winning probability of builders. We posit the following simplified expression: as a consequence of the delay induced by the timing game, the winning probability of one of the builders increases, while the other decreases. We find that the timing game will affect the probability of winning, thereby altering the time for the builder's progression towards monopoly. However, it cannot

**Table 2: Multiple Builders Fairness**

|  | No. of Builders | PBS | Collaboration | Timing Game |
|---|---|---|---|---|
| Avg. of $\lambda_i$ | 2 Builders | 0.01 | 0.00 | 0.00 |
|  | 3 Builders | 0.00 | 0.00 | 0.00 |
|  | 4 Builders | 0.00 | 0.00 | 0.00 |
|  | 10 Builders | 1.00 | 1.00 | 1.00 |

achieve robust fairness. The simulation result is shown in Appendix B.

## B  Multiple Miners/Builders

In our previous analysis, we discussed the scenario involving two block builders. We now extend this to consider the robust fairness among multiple block builders. In the PBS mechanism, the variation in connected private order flows proportions inevitably leads to one of the block builders attaining a monopoly. Therefore, the results for robust fairness in PBS align with the previous conclusions.

Table 2 displays the simulation results for multiple block builders and more rounds. The simulation involved $10^5$ blocks, with 1000 repeated experiments. Initial resource for builder $\mathcal{P}_i$ is set to $\lambda_0 = 0.2$, other parameters are the same as the previous simulation. We compared the proportion of winning probability $\lambda_i$. Clearly, robust fairness is still not realized in PBS. The builder market always tends to a monopoly.

## C  Missing Proofs

### C.1  Proof of Theorem 1

For $k \in \{i, j\}$, builder $\mathcal{P}_k$'s valuation $v_k$ has support $[\beta_k, \alpha_k]$, $0 \le \beta_k < \alpha_k$. For the sake of simplicity of the equations, it is preferable to work with the bidding price $b$ instead of $v$ to depict this equilibrium state. And we use $\phi_i(b_i)$ to denote the inverse function of $s_i(v_i, r)$. From Maskin and Riley [37, 38], there exist minimum and maximum winning bids $\underline{b}$ and $\overline{b}$ for all $b \in [\underline{b}, \overline{b}]$, and the equilibrium's differential equations can be expressed as

$$\begin{cases} \frac{f_i(\phi_i)}{F_i(\phi_i)} \phi_i'(b) = \frac{1}{\phi_j - b}, \\ \frac{f_j(\phi_j)}{F_j(\phi_j)} \phi_j'(b) = \frac{1}{\phi_i - b}. \end{cases} \quad (9)$$

The equations satisfy the following regularity condition which is

$(i)\ \phi_k(b) = s_k^{-1}(b), k \in \{i, j\},$

$(ii)\ F_k(\phi_k(\overline{b})) = 1, k \in \{i, j\},$

$(iii)\ \beta_j < r \le \beta_i \Rightarrow \underline{b} = max\{\arg\max_b\{(\beta_i - b)F_j(b)\}, r\},$

$(iv)\ r \le \beta_j < \beta_i \Rightarrow \underline{b} = \arg\max_b\{(\beta_i - b)F_j(b)\}, \phi_j(\underline{b}) = \underline{b},$

$(v)\ \beta_j < \beta_i \le r \Rightarrow \underline{b} = \phi_i(\underline{b}) = \phi_j(\underline{b}) = r.$

**Proof for $\phi_i(b) > \phi_j(b)$:** For all $x < y$ in $(\beta, \alpha_i)$, if $F_i(\cdot)$ conditional stochastically dominates $F_j(\cdot)$, we have

$$Pr(v_i < x | v_i < y) = \frac{F_i(x)}{F_i(y)} < \frac{F_j(x)}{F_j(y)} = Pr(v_j < x | v_j < y) \quad (10)$$

Rearrange and compute the first derivative,

$$\frac{F_i(x)}{F_j(x)} < \frac{F_i(y)}{F_j(y)} \quad (11)$$

$$\frac{d}{dv}\left(\frac{F_i(v)}{F_j(v)}\right) > 0, \quad v \in (\beta, \alpha_i) \tag{12}$$

which implies:

$$\frac{F_i'(v)}{F_i(v)} > \frac{F_j'(v)}{F_j(v)}, \quad v \in (\beta, \alpha_i) \tag{13}$$

Suppose that $\beta_i = \beta_j = \beta$, this implies that the lower support of the valuation distribution is influenced by the transactions contained in the public mempool. The bidding support is $[\underline{b}, \overline{b}]$, then in a punctured neighborhood of $\overline{b}$, it is obvious:

$$\alpha_i = \phi_i(\overline{b}) > \alpha_j = \phi_j(\overline{b}) \tag{14}$$

Supposed that there exists $b^* \in [\underline{b}, \overline{b}]$, and $\phi_i(b^*) = \phi_j(b^*) = v^*$, from (9), we can derive that:

$$\frac{f_i(v^*)}{F_i(v^*)}\phi_i'(b^*, r) = \frac{1}{v^* - b^*} = \frac{f_j(v_j^*)}{F_j(v^*)}\phi_j'(b^*, r) \tag{15}$$

From (13), we have $\phi_j'(b^*, r) > \phi_i'(b^*, r)$, and $\phi_j(b) > \phi_i(b)$, for $b \in [b^*, \overline{b}]$, which is contradictory with (14). Thus we have $\phi_i(b) > \phi_j(b)$.

**Proof for** $F_i(\phi_i(b)) < F_j(\phi_j(b))$ Define:

$$p_k(b) = F_k(\phi_k(b)) \tag{16}$$

and

$$H_k(\cdot) = F_k^{-1}(\cdot) \tag{17}$$

substituting (16) and (17) in to (9), we have

$$\begin{cases} \frac{p_i'}{p_i} = \frac{1}{H_j(p_j) - b}, \\ \frac{p_j'}{p_j} = \frac{1}{H_i(p_i) - b} \end{cases} \tag{18}$$

Since $F_i(\beta) = F_j(\beta) = 0, p_i(\beta) = p_j(\beta) = 0$, using L'Hôpital's Rule to (9) when $b = \beta$, we obtain:

$$\phi_i'(\beta) = \phi_j'(\beta) = 2 \tag{19}$$

By the definition of $p_k(\cdot)$, we have:

$$p_k'(b) = F_k'(\phi_k(b))\phi_k'(b), \quad k \in \{i, j\} \tag{20}$$

Combine (19) and (20), we can get:

$$p_k'(\beta) = 2F_k'(\phi_k(\beta)), \quad k \in i, j \tag{21}$$

From 13) , $F_j(\beta) > F_i(\beta)$. So it can be proved that there exist $\hat{\beta} \in [\underline{b}, \overline{b}]$ :

$$p_j(b) > p_i(b), \quad b \in [\underline{b}, \hat{\beta}]. \tag{22}$$

Suppose that we have $b^* \in [\underline{b}, \overline{b}]$ such that $\frac{p_i(b^*)}{p_j(b^*)} = 1$, put it in (18). Since $H_i(p) > H_j(p)$, for $p$ $in[0, 1]$, then we can get:

$$\frac{p_j'}{p_j} = \frac{1}{H_i(p_i) - b^*} < \frac{1}{H_j(p_j) - b^*} = \frac{p_i'}{p_i} \tag{23}$$

Then $\frac{p_i(b)}{p_j(b)}$ increases at $b^*$. For $b \in [b^*, \overline{b}]$, we always have $\frac{p_i(b)}{p_j(b)} > 1$. But we have $p_i(\overline{b}) = p_j(\overline{b})$, so $b^*$ does not exist.

In summary, we can demonstrate that (25) is valid for $b \in [\underline{b}, \overline{b}]$. That means for all bidding $b$ on the interior of their supports, we have

$$\phi_i(b) > \phi_j(b), \tag{24}$$

$$F_i(\phi_i(b)) < F_j(\phi_j(b)). \tag{25}$$

From Inequality (24), we can derive that if $s_i(v_i, r) = s_j(v_j, r)$, the MEV income $v_i$ of $\mathcal{P}_i$ surpasses $v_j$ of $\mathcal{P}_j$. Therefore, information asymmetry leads to participants with higher valuations tending to adopt more conservative bidding strategies. From Inequality (25), we can obtain $\Pr(v_i \leq \phi_i(b)) = \Pr(b_i \leq b) < \Pr(b_j \leq b) = \Pr(v_j \leq \phi_j(b))$ , with $\phi_k(b)$ strictly increasing, $k \in \{i, j\}$. Assuming the cumulative distribution function of the binding price $b$ is denoted as $p_k(b), k \in \{i, j\}$, we can obtain the bidding realization of $\mathcal{P}_i$ also first-order stochastically dominates that of $\mathcal{P}_j$.

## C.2 Proof of Theorem 2

Our analysis utilizes the techniques of Stochastic Approximation (SA) [47, 49]. We first introduce some useful definitions and lemmas of SA in the following.

**Definition 1 (Stochastic Approximation [47]).** *A stochastic approximation algorithm $\{Z_n\}$ is a stochastic process taking value in $[0, 1]$, adapted to the filtration $\mathcal{F}_n$, that satisfies,*

$$Z_{n+1} - Z_n = \gamma_{n+1}\left(f(Z_n) + U_{n+1}\right),$$

*where $\gamma_n, U_n \in \mathcal{F}_n, f \colon [0, 1] \mapsto \mathbb{R}$ and the following conditions hold almost surely*

*(1) $c_l/n \leq \gamma_n \leq c_u/n$,*
*(2) $|U_n| \leq K_u$,*
*(3) $|f(Z_n)| \leq K_f$, and*
*(4) $|\mathbb{E}[\gamma_{n+1}U_{n+1} \mid \mathcal{F}_n]| \leq K_e\gamma_n^2$,*

*where $c_l, c_u, K_u, K_f, K_e$ are finite positive real numbers.*

The stochastic approximation algorithm is originally used for root-finding problems. Specifically, $\{Z_n\}$ is a stochastic process with an initial value of $Z_0$, $\gamma_n$ denotes a moving step size gradually decreasing along with $n$ and $\gamma_n U_n$ is a random noise with an expectation tending to zero quickly. In a nutshell, $Z_n$ moves towards one of the zero points of $f(\cdot)$ and finally converges as long as the update process iterates a sufficiently large number of steps.

**Lemma 1 (Zero Point of SA [47]).** *If $f$ is continuous, then $\lim_{n\to\infty} Z_n$ exists almost surely and is in $Q_f = \{x \colon f(x) = 0\}$.*

Note that $Z_n$ may not converge to every zero point in $Q_f$. That is, if a zero point $q$ is stable, $Z_n$ converges to $q$ when $n \to \infty$ has a positive probability. Otherwise, if $q$ is an unstable point, $Z_n$ converges to $q$ with zero probability. The following lemmas characterize the properties of stable and unstable points of SA.

**Definition 2 (Attainability [47]).** *A subset $I$ is attainable if for every fixed $N \geq 0$, there exists a $n \geq N$ such that $\Pr[Z_n \in I] > 0$.*

**Lemma 2 (Stable Zero Point of SA [47]).** *Suppose $q \in Q_f$ is stable, i.e., $f(x)(x - q) < 0$ whenever $x \neq q$ is close to $q$. If every neighborhood of $q$ is attainable, then $\Pr[Z_n \to p] > 0$.*

**Lemma 3 (Unstable Zero Point of SA [47]).** *Assume that there exists an unstable point $q$ in $Q_f$, i.e., such that $f(x)(x-q) \geq 0$ locally, and that $\mathbb{E}[U_{n+1}^2 \mid \mathcal{F}_n] \geq K_L$ holds, for some $K_L > 0$, whenever $Z_n$ is close to $q$. Then, $\Pr[Z_n \to q] = 0$.*

We define $Z_n$ as the portion of the order flows received by builder $\mathcal{P}_i$ at $n$ rounds, for instance, $Z_0 = \frac{a}{a+b}$. Additionally, we consider two additional scenarios. In the first condition, private order flows

will be cut from the builder that failed in the previous round, that is, $p_n = 1$. We define $Z'_n$ as the portion of the private order flows received by builder $\mathcal{P}_i$ after $n$ rounds under this condition. Another scenario is that private order flows will be retained from the failed builder, that is, $p_n = 0$. We define $Z''_n$ as the portion of the order flows received by the builder $\mathcal{P}_i$ after $n$ rounds in this scenario.

We assume that in round $n$, the number of private order flows involving changes is $\delta_t$. Let $X_t \in \{0, 1\}$ be a binary random variable indicating whether $\mathcal{P}_i$ is the winner for the MEV-Boost auction at round $t$. Then $Z'_n$ can be written as $Z'_n = \frac{a + 2\sum_{t=1}^{n} \delta_t X_t - \sum_{t=1}^{n} \delta_t}{a+b}$. $Z''_t$ can be written as $Z''_n = \frac{a + \sum_{t=1}^{n} \delta_t X_t}{a + b + \sum_{t=1}^{n} \delta_t}$. Under any circumstances characterized by identical preceding victories or defeats, denoted as $\{Z_n \mid X_0, X_1, \ldots, X_{n-1}\}$, we can get

$$\min(Z'_n, Z''_n) \leq Z_n \leq \max(Z'_n, Z''_n). \tag{26}$$

Then, the difference between $Z''_{n+1}$ and $Z''_n$ can be written as

$$Z''_{n+1} - Z''_n = \frac{a + \sum_{t=1}^{n+1} \delta_t X_t}{a + b + \sum_{t=1}^{n+1} \delta_t} - Z''_n$$

$$= \frac{(a + b + \sum_{t=1}^{n} \delta_t) Z''_n + \delta_{t+1} X_{n+1}}{a + b + \sum_{t=1}^{n+1} \delta_t} - Z''_n$$

$$= \frac{\delta_{t+1}}{a + b + \sum_{t=1}^{n+1} \delta_t} \cdot (X_{n+1} - Z''_n).$$

Moreover, let $\gamma_{n+1} = \frac{\delta_{t+1}}{a+b+\sum_{t=1}^{n+1} \delta_t}$, $f(Z''_n) = \mathbb{E}[X_{n+1} \mid Z''_n] - Z''_n$ and $U_{n+1} = X_{n+1} - \mathbb{E}[X_{n+1} \mid Z''_n]$. Then,

$$Z''_{n+1} - Z''_n = \gamma_{n+1}(f(Z''_n) + U_{n+1}).$$

Next, we verify that conditions 1–4 given in Definition 1 hold almost surely. For condition 1, we know that $\frac{\delta_t}{(a+b+\overline{\delta_t})n} \leq \gamma_n \leq \frac{1}{n}$ and set $c_l = \delta_t/(a + b + \overline{\delta_t})$ and $c_u = 1$. For condition 2, we set $K_u = 1$ as $|U_n| \leq 1$. For condition 3, we know that

$$f(Z''_n) = \begin{cases} \frac{Z''_n}{2(1-Z''_n)} - Z''_n, & \text{if } Z''_n \leq \frac{1}{2}, \\ 1 - \frac{1-Z''_n}{2Z''_n} - Z''_n, & \text{otherwise.} \end{cases} \tag{27}$$

Thus, it can be seen that $\left|f(Z''_n)\right| \leq 1$ and hence we set $K_f = 1$. Finally, for condition 4, we find that $\mathbb{E}[\gamma_{n+1} U_{n+1} \mid \mathcal{F}_n] = 0$ and hence we set $K_e = 0$.

In addition, by Equation (27), we observe that $f(Z''_n)$ is continuous for $Z''_n \in [0, 1]$. Thus, by Lemma 1, $\lim_{n\to\infty} Z''_n$ exists almost surely and is in one of the zeros of $f(\cdot)$. Let $f(x) = 0$ such that the zeros are found as $Q_f = \{0, \frac{1}{2}, 1\}$. Then, it remains to show that $q = 1/2$ is an unstable point and $q = 0$ and $q = 1$ are two stable points.

Clearly, we have

$$f(x)(x - 1/2) = \begin{cases} \frac{x(x-1/2)}{1-x} \cdot (x - 1/2) \geq 0, & \text{if } x \leq \frac{1}{2}, \\ \frac{(1-x)(x-1/2)}{x} \cdot (x - 1/2) \geq 0, & \text{otherwise.} \end{cases}$$

Furthermore,

$$\mathbb{E}[U_{n+1}^2 \mid \mathcal{F}_n] = \mathbb{E}[X_{n+1}^2 \mid Z''_n] - \mathbb{E}^2[X_{n+1} \mid Z''_n]$$
$$= \mathbb{E}[X_{n+1} \mid Z''_n] - \mathbb{E}^2[X_{n+1} \mid Z''_n].$$

Thus, if $Z''_n$ is close to 1/2, i.e., $Z''_n \in [1/2 - \varepsilon, 1/2 + \varepsilon]$ for some $\varepsilon > 0$, we have $\mathbb{E}[X_{n+1} \mid Z''_n] \in [\frac{1/2-\varepsilon}{1+2\varepsilon}, \frac{1/2+3\varepsilon}{1+2\varepsilon}]$. As a result,

$$\mathbb{E}[U_{n+1}^2 \mid \mathcal{F}_n] \geq \frac{1/2 - \varepsilon}{1 + 2\varepsilon} \cdot \frac{1/2 + 3\varepsilon}{1 + 2\varepsilon} \triangleq K_L,$$

which implies $q = 1/2$ is an unstable point. Hence, according to Lemma 3, $\Pr[Z''_n n \to 1/2] = 0$.

Finally, we prove that $q = 0$ is a stable point, with $q = 1$ being analogous. Using a similar method, we can also prove that for $Z'_n$, there are only two stable points at $q = 0$ and $q = 1$. Therefore, we can get the conclusion that $Z_n$ will approach 0 or 1 when $n \to \infty$. Note that when $Z_n \to 0$, we must have $\lambda_i \to 0$. Therefore, when $n \to \infty$, $\Pr[(1-\varepsilon)\lambda_0 \leq \lambda_i \leq (1+\varepsilon)\lambda_0] = 0$ for any positive $\varepsilon$, which concludes the theorem.

## C.3 Proof of Theorem 3

**Proof for** $F_s(\phi_s(b)) < F_a(\phi_a(b))$ **and** $\phi_a(b) < \phi_s(b)$: We define $[\underline{u}, \overline{u}]$ as the support of bidding valuation in the asymmetric situation and $[\underline{b}_s, \overline{b}_s]$ represent the bidding support of the symmetric auction with bidders' valuations. We can derive that $\overline{b}_s \leq \overline{u}$, hence we have $\phi_a(b) \geq \phi_a(b)$. For any $b \in (\beta_s, \overline{b})$, such that $\phi_a(b) \leq \phi_a(b)$.

$$\frac{F'_a(\phi_a)}{F_a(\phi_a)} \phi'_a = \frac{1}{\phi_j - b} > \frac{1}{\phi_i - b} \geq \frac{1}{\phi_a - b} = \frac{F'_s(\phi_s)}{F_s(\phi_s)} \phi'_s. \tag{28}$$

Hence,

$$\phi_a(b) \leq \phi_s(b). \tag{29}$$

$$\frac{d}{dv}\left(\frac{F_a(\phi_a)}{F_s(\phi_s)}\right) > 0. \tag{30}$$

For some $\hat{\theta} \leq 1$, suppose that there exits $\hat{b} \in (\beta_s, b^*)$ satisfying

$$\frac{F_a(\phi_a(\hat{b}))}{F_s(\phi_s(\hat{b}))} = \hat{\theta}. \tag{31}$$

Because $\frac{F_a(\phi_a(\hat{b}))}{F_s(\phi_s(\hat{b}))}$ is increasing at $\hat{b}$, thus we have

$$\phi_a(b) < \phi_s(b) \quad and \quad \frac{F_a(\phi_a(\hat{b}))}{F_s(\phi_s(\hat{b}))} > 0. \tag{32}$$

But $\phi_s(b_s) = b_s$ and so $\phi_a(b_s) \geq \phi_s(b_s)$, which is contradiction with Formula 29

**Proof for** $R^I > R^{II}$ **with**

$$R^I = \int b \, d\big(H(\phi_1(b))^m H(\phi_2(b))^n\big),$$

$$R^{II} = \int b \, d\big(H(\widetilde{\phi}_1(b))^{m'} H(\widetilde{\phi}_2(b))^{n'}\big).$$

Denote by $(\phi_1, \phi_2)$ the equilibrium in the scenario I, and by $(\widetilde{\phi}_1, \widetilde{\phi}_2)$ the equilibrium in the scenario II. Let $G_I(b) = H(\phi_1(b))^m H(\phi_2(b))^n$ and $G_{II}(b) = H(\widetilde{\phi}_1(b))^{m'} H(\widetilde{\phi}_2(b))^{n'}$, that is $G_I$ and $G_{II}$ are the cumulative distribution of bids under scenario I and scenario II respectively. With these notions,

$$R^I(b) = \int b \, dG_I(b). \tag{33}$$

### Table 3: Statistic data of top builders

| Builders | Blocks [#] | Market Share [%] | Total Payments [ETH] | Total Block Value [ETH] | Profit Margin [%] |
|---|---|---|---|---|---|
| beaverbuild | 44,602 | 50.5 | 6278.51 | 8036.49 | 13.58 |
| Titan | 19,927 | 22.56 | 3479.22 | 6682.89 | 8.34 |
| rsync-builder | 12,043 | 13.64 | 1651.13 | 2288.90 | 27.66 |
| jetbldr | 2,731 | 3.09 | 189.31 | 188.27 | -9.91 |
| Flashbots: Builder | 2,556 | 2.89 | 472.00 | 573.89 | 12.34 |
| f1b | 1,572 | 1.80 | 131.17 | 158.94 | 13.48 |
| tbuilder | 1,339 | 1.52 | 52.36 | 55.60 | -5.77 |
| builder0x69 | 1,312 | 1.49 | 215.43 | 327.18 | 16.68 |
| penguinbuild | 708 | 0.80 | 101.64 | 124.74 | 14.47 |
| lokibuilder | 359 | 0.41 | 24.92 | 31.37 | 13.47 |

$$R^{II}(b) = \int b \, dG_{II}(b). \tag{34}$$

A sufficient condition for $R^I(b) > R^{II}(b)$ is that $G_I(b) < G_{II}(b)$ in the interior of their common support. Using first-order stochastic dominance, we have

$$G_I(b) = G_{II}(b) \Rightarrow \frac{G'_I(b)}{G_I(b)} < \frac{G'_{II}(b)}{G_{II}(b)}. \tag{35}$$

This allows us to rule out any crossing of $G_I(b)$ and $G_{II}(b)$ to the right of $\underline{b}$, $\underline{b}$ is the lower bounder of bidding value and using Formula 35 we can easily derive $R^I(b) > R^{II}(b)$.

## D Builder Profit

In Table 3, in addition to builder's market share and profit margin, we also calculate the total payments to proposers and total block value of builders.

xiv