# OpenReview forum: "Private Order Flows and Builder Bidding Dynamics: The Road to Monopoly in Ethereum’s Block Building Market"
_ACM.org/TheWebConf/2025/Conference — WWW 2025 Poster_

### Official Review · Reviewer_e9xc · 2024-10-31

**Novelty:** 2
**Technical Quality:** 2

**Review:**

# Review of Paper
The research topic is both relevant and timely, addressing competitive dynamics and auction strategies within the context of private order flows in the MEV-Boost auction. However, while the topic has potential, the manuscript requires substantial revisions in quality, clarity, and empirical rigor to meet publication standards.

The manuscript lacks empirical evidence to substantiate its primary claims about private order flows and builder profitability. Additionally, inconsistencies in notation, ambiguities in assumptions, and structural issues affect overall coherence. Ambiguous definitions (e.g., “weak builder”), unclear notation (e.g., using $a$ and $b$ for private flows without time dependency), and transitions that fail to logically connect theoretical assumptions with empirical observations detract from clarity. Reorganization and additional proofreading would significantly improve readability.
## Pros and Cons
### Pros
- **Relevant Topic**: The analysis of private order flows in Ethereum’s block-building market is both timely and pertinent to ongoing research in blockchain economics.
- **Novelty in Auction Strategy Analysis**: The paper introduces a theoretical framework to assess the competitive dynamics between builders with different access to private order flows.

### Cons
- **Ambiguities and Inconsistencies**: Several terms and notations, such as "weak builder" and the static definitions of $a$ and $b$, require clarification. This ambiguity makes it challenging to interpret the results meaningfully.
- **Structure and Readability**: The paper would benefit from a clearer structure, linking theoretical assumptions with empirical observations. Reorganization is recommended, particularly in sections discussing assumptions and empirical results.
- **Unclear Figures**: Figures 2 and 3 are ambiguous, with unclear axes and labels, making it difficult to interpret the data presented. Additional captions and a clearer depiction of ratios and relationships would help.

## Recommendations for Revision
1. **Empirical Validation**: To substantiate claims regarding private order flows and builder revenue, it is essential to include data comparing the profitability of blocks constructed from private versus public order flows. This data would provide necessary support for the claim that private flows are advantageous to builders.

2. **Notation and Assumptions**: Reconsider the notation for private order flows, specifically the static definitions of $a$ and $b$. Introducing time-varying variables, such as $a_t$ and $b_t$, would more accurately reflect the stochastic dynamics described in the model assumptions.

3. **Improving Clarity and Readability**: Address inconsistencies in terminology, such as "weak builder," and improve the structure of the paper to ensure a logical flow. For instance, Section 3.2 could benefit from linking the discussion of the rsync-builder more directly to the assumptions in the binomial model.

4. **Clarifying Figures**: Enhance Figures 2 and 3 by providing detailed captions and clearer labels for axes and variables. This adjustment will improve the interpretability of the empirical evidence.

**Questions:**

**Question #1**
In Section 3.2, line 328, the term 'weak builder' is ambiguous and requires a more explicit definition. It might refer to builders who primarily utilize transactions from the public mempool, lacking access to private order flows, but this should be clearly stated to avoid confusion.

Additionally, the claim that private order flows provide builders with higher profits than public mempool transactions requires further clarification. In general, transaction inclusion is driven by transaction fees, with higher-fee transactions being prioritized regardless of whether they are routed through private or public channels. The current explanation seems to overlook this priority and leaves it unclear why private transactions would inherently yield higher profits.

If searchers or users aim to perform high-stakes MEV activities, such as sandwich attacks or arbitrage with flash loans, they may indeed offer higher transaction fees to ensure timely inclusion. However, such transactions would likely be prioritized and included in a block regardless of whether they are sent through public or private channels. It’s also possible that certain arbitrage traders prefer private flows to avoid detection and maintain a competitive advantage. This preference would likely increase the share of private transactions directed toward established builders, but it’s unclear how much this actually impacts total block rewards and building profits.

**Question #2**
Lack of clarity on the figures’ purpose. Authors should describe more details in captions to enhance readability.

**Question #3**
In section 3.2, the authors state that private order flows contribute 54.59% of block rewards, implying that access to private flows directly enables higher block rewards and builder dominance. However, it’s more accurate to say that private order flows are a byproduct of high market share rather than the primary cause of it. High-market-share builders naturally receive more private transactions simply because they win more blocks. High market share appears to be a necessary condition for receiving substantial private order flows. To clarify this relationship, it would help if the authors discussed whether private order flows alone are sufficient to drive builder centralization, or if they are simply amplifying an existing market share advantage.

**Question #4**
Figures 2 and 3 are somewhat ambiguous in terms of their axes and the data they represent. In Figure 2, the x-axis is labeled 'Proportion of Private Rewards,' while the y-axis is 'Number of Transactions,' but it’s unclear how transaction counts relate directly to reward proportions. It seems the authors may be attempting to convey that blocks with a certain number of transactions yield a proportion of rewards from private sources. However, this relationship is not explicitly shown, and it’s difficult to interpret the exact meaning. Similarly, in Figure 3, the x-axis ('Proportion of Private Transaction Counts') and y-axis ('Number of Transactions') lack clarity in depicting whether this represents an overall ratio of private to public transactions for each builder.

Additionally, there’s no clear depiction of the ratio of private to public transactions per block or per builder, which would provide a more straightforward view of how private transactions impact rewards. Finally, the purpose of the solid line denoting 'counts of private transactions' is unclear in its relationship to the histogram distribution. A more informative representation might involve plotting average rewards per block against the proportion of private transactions, as this would directly illustrate how private transactions impact builder rewards.

**Question #5**
On page 4 line 356, to avoid ambiguity, it would be helpful to define $a$ and $b$ as the cardinalities (sizes) of the sets of private order flows associated with builders $P_i$ and $P_j$, respectively. For example, using $|A|$ and $|B|$ for these sizes would make expressions like $|A| + |B|$ clearer and avoid confusion around set membership. This notation would also more accurately convey the intended proportion of private transactions controlled by each builder.

The phrase "we recomputed the overlapping private order flows and utilized $a + b$ as the total counts within the market" is somewhat unclear. If "overlapping" refers to transactions that appear in both sets $a$ and $b$ (i.e., the intersection $A \cap B$), then using $a + b$ as the total count could lead to double-counting these shared transactions. To avoid this, it might be more accurate to use $|A \cup B| = |A| + |B| - |A \cap B|$ as the total count of unique private order flows across the market.

**Question #6**
On Page 4, line 360, the phrase, "The expectation counts of the private order flows of each builder’s respective block satisfy a binomial distribution," is confusing and may need reconsideration regarding the underlying assumptions. Specifically, it does not appear to account for market share differences between builders. For instance, assuming two builders, the proportion of private order flows each builder receives should logically depend on their respective market shares. If the market shares are $x$ and $1 - x$ for the two builders, these proportions should be incorporated into a binomial distribution model.

Additionally, if the intention is to model the distribution based on the number of private order flows accessible to each builder, then the total number of private transactions available to both builders must be specified. For example, let $A$ represent the private order flows accessible to builder $i$, $B$ represent the private order flows accessible to builder $j$, and $C$ represent the intersection $A \cap B$, or the private flows accessible to both builders. Then, the proportion of accessible private order flows for each builder would be $\frac{|A|}{|A| + |B| - |C|}$ for builder $i$ and $\frac{|B|}{|A| + |B| - |C|}$ for builder $j$. Currently, the definition of this distribution is unclear in the text.

**Question #7**
To enhance readability for audiences outside the blockchain community, consider providing brief definitions or introductions for terms like Erigon node, Lightnode, and rsync-builder, as these may not be familiar to all readers

**Question #8**
Section 3.2 is an inconsistency in the flow between these paragraphs. After introducing the binomial distribution model and assumptions about private order flows, the narrative shifts abruptly to discussing the growth of the rsync-builder without linking it to the previous assumptions. To improve cohesion, consider framing the rsync-builder as an empirical example to validate or test the binomial model assumptions. Clearly connect the observations about rsync-builder’s private bundles and market share changes to the earlier theoretical framework, so readers can see how the empirical data relate to the initial assumptions.

Furthermore, the variable $\delta$ is used in Section 2.2 'Robust Fairness' and $\delta_t$ in Assumption 1. If these are intended to represent the same variable, it would be helpful to include a reminder or clarification for consistency. If they represent different variables, it would be clearer to avoid reusing the same notation to prevent confusion.

**Question #9**
Assumption 1 defines $Z_i^t$ as a time-varying proportion of private order flows, but the initial definitions of $a$ and $b$ do not suggest they are time-dependent. To reflect the stochastic process dynamics described in Assumption 1, it would be more rigorous to redefine $a$ and $b$ as time-varying variables, such as $a_t$ and $b_t$. This way, $Z_i^t$ could be represented as $Z_i^t = \frac{a_t}{a_t + b_t}$, allowing the model to reflect the changes in private order flows as each round progresses. Without this time dependency, the stochastic assumption lacks clarity and may not accurately represent the intended dynamics of private order flows.

Additionally, the current model describes a binomial distribution(or process), which implies a fixed probability of success for each builder across rounds. However, I am uncertain if a binomial distribution is suitable for modeling this situation. Since Assumption 1 describes a time-varying proportion of private order flows connected to each builder, a binomial model with constant probabilities may be inappropriate. Either Assumption 1 or the distribution model itself may need to be revised to account for these time-varying probabilities.

**Question #10**
The section 3.3 Auction Strategy Difference is challenging to follow, particularly due to the dense mathematical formulation without clear, intuitive explanations. The purpose of each equation (particularly (2), (3), and (6)) would be clearer if accompanied by a more detailed explanation of each variable and function (e.g., defining $g$ in Equation (3) and explaining its role in valuation). Additionally, tying the model assumptions about private flows to empirical insights or real-world implications would make this section more accessible and relevant to readers. For readability, consider reorganizing to flow logically from assumptions to model derivation to empirical validation, clearly linking each part of the analysis to Theorem 1.

**Reviewer Confidence:**

4: The reviewer is certain that the evaluation is correct and very familiar with the relevant literature

**Scope:**

3: The work is somewhat relevant to the Web and to the track, and is of narrow interest to a sub-community

---

### Official Review · Reviewer_c9LH · 2024-12-01

**Novelty:** 6
**Technical Quality:** 6

**Review:**

This paper studies how private order flows in Ethereum’s Proposer-Builder Separation (PBS) drive centralization in the block-building market. The authors introduce information differences (block valuation disparities) and auction strategy differences (asymmetric bidding), showing that dominant builders with access to private transactions win most auctions. The study uses MEV-Boost data to verify how this concentration reduces proposer revenue, delays low-fee transactions, and increases censorship risks, undermining Ethereum’s decentralization.

The main contribution of this work is identifying two forms of differences in the builder market information difference and auction strategy differences and validating our theoretical analysis through bidding data from builders. The authors also analyze the impact of information differences and auction strategy differences.

I have to say that I am not the right person to judge the technical novelty of this paper, as I usually work on approximation and online algorithms. To me, the paper seems to study an interesting problem, and this problem is also well-motivated.

**Questions:**

I don't have any specific questions.

**Ethics Review Flag:**

Yes

**Reviewer Confidence:**

2: The reviewer is willing to defend the evaluation, but it is likely that the reviewer did not understand parts of the paper

**Scope:**

4: The work is relevant to the Web and to the track, and is of broad interest to the community

---

### Official Review · Reviewer_kceh · 2024-12-02

**Novelty:** 5
**Technical Quality:** 5

**Review:**

Summary

This paper studies the implications of private order flows in Ethereum.  It is observed that builders receiving more private orders have higher valuation for a block; they tend to bid more aggressively in the auction in which the winner's block gets to be selected by Ethereum.  Both these effects push up the winning probability of such a builder, which in turn attracts more private orders.  This is theoretically modelled and verified using empirical data.  The market therefore tends to evolves to a quasi monopoly, where the one or fewer strongest builders win the vast majority of the auctions.  This beats the purpose of the Proposer Builder Separation (PBS) system which aims to prevent the concentration of block profits.  This concentration hurts the profits of most market participants, and also exacerbates the congestion, as the stronger builders take more private orders and fewer transactions from the public pool.


Evaluations


Strength

This is a timely study of a phenomenon in a real online market.  It makes plausible diagnosis and corroborates is through empirical data.


Weakness

The economics/dynamics underlying the phenomenon seems a rather simple positive feedback loop.  When a market gives positive feedback instead of negative feedback, competition tends to diminish and the winner in the early rounds tends to take over all the market.  This is well known, and seems to be just reconfirmed here.  (Added after rebuttal: the authors claim this is nontrivial, but the reason given is that the mechanism was introduced to counter another existing positive feedback loop.  I don't find that altogether convincing.) Even though the theoretical analysis here uses some sophisticated math, the conclusion seems rather expected.  In this sense, the paper has less theoretical interest, with its main merit being observing, and providing evidence for, that this is happening in Ethereum.

**Questions:**

Theorem 1 seems not a rigorous statement, or uses a language that is not well defined.  What does "bids less aggressively" mean exactly?

**Reviewer Confidence:**

2: The reviewer is willing to defend the evaluation, but it is likely that the reviewer did not understand parts of the paper

**Scope:**

3: The work is somewhat relevant to the Web and to the track, and is of narrow interest to a sub-community

---

### Official Review · Reviewer_7Pxv · 2024-12-02

**Novelty:** 6
**Technical Quality:** 6

**Review:**

### **Review Comments**

#### **Quality**

This paper provides an in-depth theoretical and empirical analysis of the **Proposer-Builder Separation** (PBS) mechanism in Ethereum, particularly focusing on the impact of private order flows on the builder market. The paper clearly presents the theoretical framework and supports the analysis with robust experimental data. The **empirical analysis** effectively demonstrates the behavior of different builders, private transaction flows, and market dynamics under PBS, providing strong data support for the study. The theoretical framework is well-grounded, the data analysis is clear, and the methodology is scientifically sound.

#### **Clarity**

The structure of the paper is generally clear, and the chapters are logically connected. However, the background section is somewhat lengthy, and I suggest breaking it down into smaller sections to ensure that readers can easily understand, especially when encountering concepts like PBS mechanisms and private order flows for the first time. Additionally, **some formulas in Section 5.1** are not numbered, which reduces clarity and may confuse readers when referring to them. Improving these details will greatly enhance the **readability** of the paper.

There is also room for improvement in **figures**, particularly **Figure 7**, which has limited information and lacks sufficient analysis. Adding more layers of information and comparisons will improve the **effectiveness** of the figures.

#### **Originality**

The originality of this paper lies in its novel perspective on **private order flows**, which has been incorporated to analyze the dynamics of the builder market under the PBS framework. The paper introduces the concepts of **information difference** and **auction strategy difference**, revealing the potential market centralization and monopolistic trends within the current PBS mechanism. While PBS has been previously discussed in Ethereum research, this paper provides a new angle on how private order flows can lead to bidding strategy differences among builders, and eventually drive the market toward monopoly, thus offering an important addition to blockchain fairness research.

The combination of data and theoretical models explores the flaws of the PBS mechanism and its long-term impact, providing **innovative theoretical contributions**.

#### **Significance**

The significance of this paper lies in its thorough investigation into the **potential limitations** of the PBS mechanism, particularly in the context of private order flows, which could lead to gradual monopolization of the builder market. The authors combine theoretical deductions with empirical data to show how private order flows negatively impact market fairness and decentralization, suggesting necessary optimizations for the PBS mechanism. This research holds significant **policy implications** and **practical guidance** for Ethereum and other blockchain developers, offering valuable insights into the current blockchain reward distribution mechanisms.

The study also highlights the **complexity** of achieving **fairness** and **decentralization** in blockchain networks, showing how **information asymmetry** and **auction strategy differences** can lead to market concentration and affect the profits of stakers and other network participants. These results are of broad **influence** for designing more equitable blockchain protocols.

---

### **Strengths**

1. **Clear Structure**: The theoretical framework and data analysis structure are clear and effectively support the paper's central argument.
2. **Novel Perspective**: The introduction of private order flows provides a fresh perspective on PBS mechanism analysis.
3. **Comprehensive Empirical Analysis**: The empirical section is well-developed, effectively validating the theoretical hypotheses, and the results strongly support the paper's claims.

### **Weaknesses**

1. **Figure 7 has limited information**: The figure is too simplistic and lacks sufficient analysis and comparison. More layers of detail and comparison would improve its effectiveness.
2. **Background Section is Imbalanced**: The background section is too lengthy, and it would benefit from being broken down into more sections to improve readability.
3. **Formulas in Section 5.1 are not numbered**: Some formulas lack numbering, reducing the paper's academic rigor. It is recommended to number all formulas, particularly in **Section 5.1**.

**Questions:**

### **Questions for the Authors**

1. Is this phenomenon specific to the early stages of PBS, or do you anticipate it will become more pronounced in the future as the system matures?

2. Are there any simple solutions to mitigate this issue, or would it require more complex mechanisms to address the monopolistic trends in the builder market?

3. How long did it take to complete the extensive experiments presented in the paper? Could you share any challenges or insights from the data collection and analysis process?

**Reviewer Confidence:**

4: The reviewer is certain that the evaluation is correct and very familiar with the relevant literature

**Scope:**

4: The work is relevant to the Web and to the track, and is of broad interest to the community

---

### Official Review · Reviewer_1LRe · 2024-12-03

**Novelty:** 4
**Technical Quality:** 3

**Review:**

This paper disccuses the Ethereum block-building market and the impact of private order flows. Private order flows, which constitute transactions and bundles sent directly to builders, have significantly altered the dynamics of the block building market. They show that although they make up only 12% of all transactions, they contribute to 54.59% of block rewards. This phenomenon is termed "information difference" because different builders have access to different information about transactions and therefore different valuations of the blocks they are building. The paper explores the impact of information difference on the MEV-Boost auction, the process by which builders compete to have their blocks selected. They argue that this information asymmetry leads to "auction strategy difference", where builders with higher valuations (due to more private order flows) bid less aggressively than those with lower valuations. This is because they are more confident of winning the auction even with lower bids. Finally, they discuss the negative implications of this monopolistic trend, including reduced revenue for proposers and increased discrimination in block construction, with transactions offering lower priority fees experiencing longer delays.

A few comments:

-- I'm not sure what to make of this paper, I do like it but several papers (notably Gupta, Pai and Resnick, but also Oz et al) have made similar points, and those papers are either older (the former) or have significantly more in-depth and recent data (the latter). In particular, the market landscape has already changed substantially since the end of the dataset considered here.

-- It is not clear to me how to interpret Assumption 1. Private order flow in Ethereum is mostly arranged by long-term contracts.

-- I don't understand the robust fairness criterion. What is the block builder's "initial investment"? A lot of these order-flow arrangements include kicking back some portion of the profits from the private order flow to the orderflow provider, including via order flow auctions (MEVBlocker, Blink etc), or long term contracts (telegram bots).

**Questions:**

See Above

**Reviewer Confidence:**

4: The reviewer is certain that the evaluation is correct and very familiar with the relevant literature

**Scope:**

3: The work is somewhat relevant to the Web and to the track, and is of narrow interest to a sub-community